# Hyperspectral Imaging and Machine Learning as a Nondestructive Method for Proso Millet Seed Detection and Classification

**DOI:** 10.3390/foods13091330

**Published:** 2024-04-26

**Authors:** Nader Ekramirad, Lauren Doyle, Julia Loeb, Dipak Santra, Akinbode A. Adedeji

**Affiliations:** 1Department of Biosystems and Agricultural Engineering, University of Kentucky, Lexington, KY 40546, USA; n.ekramirad@gmail.com (N.E.); laurenedoyle@uky.edu (L.D.); julia.loeb@uky.edu (J.L.); 2Panhandle Research and Extension Center, 4502 Avenue I, Scottsbluff, NE 69361, USA; dsantra2@unl.edu

**Keywords:** millet, proso millet variety, hyperspectral imaging, near infrared, machine learning

## Abstract

Millet is a small-seeded cereal crop with big potential. There are many different cultivars of proso millet (*Panicum miliaceum* L.) with different characteristics, bringing forth the issue of sorting which are important for growers, processors, and consumers. Current methods of grain cultivar detection and classification are subjective, destructive, and time-consuming. Therefore, there is a need to develop nondestructive methods for sorting the cultivars of proso millet. In this study, the feasibility of using near-infrared (NIR) hyperspectral imaging (900–1700 nm) to discriminate between different cultivars of proso millet seeds was evaluated. A total of 5000 proso millet seeds were randomly obtained and investigated from the ten most popular cultivars in the United States, namely Cerise, Cope, Earlybird, Huntsman, Minco, Plateau, Rise, Snowbird, Sunrise, and Sunup. To reduce the large dimensionality of the hyperspectral imaging, principal component analysis (PCA) was applied, and the first two principal components were used as spectral features for building the classification models because they had the largest variance. The classification performance showed prediction accuracy rates as high as 99% for classifying the different cultivars of proso millet using a Gradient tree boosting ensemble machine learning algorithm. Moreover, the classification was successfully performed using only 15 and 5 selected spectral features (wavelengths), with an accuracy of 98.14% and 97.6%, respectively. The overall results indicate that NIR hyperspectral imaging could be used as a rapid and nondestructive method for the classification of proso millet seeds.

## 1. Introduction

Millet is a small-seeded crop compared to other cereal grains with great potential owing to its valuable qualities such as containing comparable levels of essential nutrients [1], drought resistance [2], being gluten free [3], having a short maturity period [4,5], having low water requirements for cultivation [6], and having the ability to grow well on marginal land such as acidic soil [7]. These qualities make millet a great crop to be grown in arid areas of Africa and Asia. It is considered a crop that can help to address the problem of food insecurity [8].

There are nearly 20 different species of millet grown worldwide [9], known by different names such as finger millet, foxtail millet, pearl millet, little millet, sorghum millet and proso millet. In the United States, proso millet (*Panicum miliaceum* L.) is the most cultivated variety of millet, with a harvest of 705,000 ha in 2023 [10]. Proso millet is a nutritionally rich cereal grain that is rarely used as human food in the US but mostly as animal (bird) feed, as well as planted for forage and ethanol production [11]. Proso millet has many different cultivars, with some farmers producing more than one cultivar on a single plot of land, which may lead to the cross-contamination and sorting of the cultivated seeds. Being able to differentiate between the variety of proso millet is very critical, especially given that some cultivars have distinct macromolecule and proximate compositions that are significantly different from those of the rest [11]. Singh et al. [12] reported that the starch qualities of 10 cultivars of proso millet grown in the US are significantly different. The levels of amylose to amylopectin reported in the proso millet cultivars showed differences that could impact their functionality as an ingredient in food production. The quality of food products depends on the functionality characteristics of the raw ingredients. Therefore, clear distinctions in food composition, purity, and physicochemical characteristics are very important for the food processors, plant breeders, farmers, and scientific researchers [13]. Current methods of seed classification use visible morphological and phenotypical identification methods, DNA molecular marking technology, and protein electrophoresis, among others [14,15]. Morphological methods are highly laborious and subjective. All other techniques suffer from shortcomings such as being destructive, being time-consuming, needing trained personnel, creating wastes with attending negative environment impacts, and not being suitable for large-scale application. Therefore, there is a need to develop effective nondestructive methods for the classification of millet seed cultivars with application in cultivar contamination detection that could occur at several points along the cereal’s supply chain—during harvesting and grain processing.

There are several methods developed for the nondestructive detection and evaluation of food materials such as spectroscopy [16,17], spectral vegetation indices [18], machine visions [19], and hyperspectral imaging [14]. Among these, hyperspectral imaging (HSI) can provide the most information, as it combines the techniques of machine vision and spectroscopy. Machine vision is an automated computer system that visually inspects the sample, operating at visible wavelengths in a range suitable to give spatial information about the external view and positions of certain features [20]. This, however, cannot provide an internal view of the object, missing the ability to detect complex classifications and quality compositions [20]. On the other hand, spectroscopy obtains the spectral data of a sample based on light absorbance or reflectance. This gives information about the internal components of a sample but does not pinpoint its exact location [20]. HSI combines the power of machine visions and spectroscopy and more by providing the spectrum for each pixel in an object’s image, creating what is called a hypercube with spatial dimensions on two axes, and a wavelength spectral dimension in the third axis [21]. With this added dimension, HSI can provide a spectrum for every pixel as a chemical map, giving it the advantage of elucidating more information about the sample [21]. This method works especially well for samples whose nature is not homogeneous, such as seeds that have an uneven distribution of internal chemical components [15].

Most of the research on the application of HSI as a nondestructive method for grain classification focuses on larger grain types such as soybean [15,22], wheat [23], oat [24], corn [25], and rice [26], giving less attention to the small-grain cereals such as millet. Only a limited number of studies were found in the literature for the use of hyperspectral imaging to evaluate small grain seeds such as millet. For example, Wang et al. [27] developed an effective and fast technique to identify the origin of foxtail millet seeds using HSI in the 900 to 1700 nm range. They obtained a prediction set accuracy of 95% using the principal component analysis and support vector machine (PCA-SVM) classification model. They concluded that further studies should be undertaken to investigate more representative millet samples from more regions to build an extended database and introduce new features to develop more general discriminative models. In that line, Wang et al. [14] used visible near-infrared (Vis-NIR) hyperspectral imaging to classify 480 samples of eight millet varieties with an attention-convoluted recurrent neural network (attention-CRNN) model, giving a test set accuracy of 87.5%. This study had a relatively low classification performance and used less common millet cultivars. They concluded that further improvement of the models is needed to obtain higher accuracies with fewer wavelengths for practical applications and considered that different millet varieties usually have more distinct differences than cultivars within a variety. To improve their results, there may be a need to increase the number of seed replicates, use the NIR spectra range, and explore additional machine learning models. Besides the work carried out by Wang et al. [14], there is no known work on proso millet cultivar classification using HSI. However, this topic has been covered by others using different cereals. Zhang et al. [11] developed a model to discriminate 330 samples of maize seeds of six varieties using Vis-NIR HSI, reaching 98.9% accuracy. In another study on maize seeds conducted by Zhao et al. [28], NIR HSI was applied to scan 12,900 seeds of three different varieties to classify them into three different varieties using a machine learning classifier. Another study on 17 varieties of maize seeds conducted by Xia et al. [28] that used a Vis-NIR HSI to scan 1623 seeds with a multi-linear discriminate analysis (MLDA) algorithm resulted in a 99.1% accuracy. Kong et al. [29] identified four different rice cultivars with 225 seeds with a classification rate over 80% using a NIR HSI system. Zhou et al. [30] introduced a method for identifying nine varieties of sweet maize seeds using 90 samples for each variety via Vis-NIR HSI (326.7–1098.1 nm), with the best accuracy of 94% recorded for the combination of Savitzky–Golay and first-derivative preprocessing, the competitive adaptive reweighted sampling (CARS) wavelength selection algorithm, and the SVM classification model. They suggested using more varieties, as well as more seeds, to build more reliable models in future works. These studies show the potential of applying NIR HSI for discriminating between varieties of large cereal grains with distinct features.

Modern hyperspectral imaging techniques have great potential in the analysis of food products by providing detailed spectral information across a wide range of wavelengths. Unlike traditional color imaging methods, hyperspectral imaging captures spectral signatures at hundreds of contiguous narrow wavelength bands, enabling the detection and quantification of various chemical compounds present in the millet seeds, including, but not limited, to proteins, lipids, carbohydrates, vitamins, minerals, and phytochemicals. Each of these constituents exhibits a unique spectral signature, allowing for their differentiation and quantification using hyperspectral data analysis techniques. Machine learning algorithms play a vital role in interpreting hyperspectral data and extracting meaningful information regarding the chemical compositions of millet seeds by correlating the spectral features with specific chemical constituents. By training these models on spectral data paired with the labels of millet classes (varieties), classification models can be built to discriminate different seeds in the test dataset. Therefore, the main objective of this study was to explore the feasibility of using shortwave near-infrared (SWNIR) HSI for the classification of different cultivars of proso millet seeds. The specific objectives of this study were to (1) determine the spectral profile of each proso millet cultivar using hyperspectral imaging, (2) use this dataset to create a supervised algorithm to qualitatively classify the millet cultivars, and (3) develop multispectral HSI models using the optimum wavelengths.

## 2. Materials and Methods

### 2.1. Sample Preparation

In this study, ten different cultivars of proso millet seeds were grown and supplied by the University of Nebraska’s Panhandle Research and Extension Center in Scottsbluff, Nebraska, U.S.A. The ten cultivars used were Cerise, Cope, Earlybird, Huntsman, Minco, Plateau, Rise, Snowbird, Sunrise, and Sunup. The seeds were sifted to remove any outside contaminates like stones, dirt, and straw using a Ro-Tap sieve shaker (RX-29, W.S. Tyler, Mentor, OH, USA), then dehulled with a modified disc attrition mill (Glenn Mills Inc., Clifton, NJ, USA), where a rubber disk head was attached to the stationary part to minimize the breakage of the millet. The seeds were stored in a deep-freezer with a temperature of −70 °C in properly labeled bags. To be prepared for scanning, the seeds were taken out of the freezer and placed at room temperature (≃25 °C) for at least one hour before testing. A total of 500 seeds were randomly counted out from each bag and separated into 10 groups of 50 seeds. The hyperspectral data acquisition step was carried out in the Food Engineering Laboratory at the Biosystems and Agricultural Engineering Department, the University of Kentucky, Lexington, KY, USA.

### 2.2. Hyperspectral Imaging Data Acquisition Procedure

An HSI system based on shortwave near-infrared (NIR) bands was used to acquire the spatial–spectral data, as shown in Figure 1. The HSI system consisted of a NIR spectrograph with a wavelength range from 900 nm to 1700 nm and a spectral resolution of 3 nm (N17E, Specim, Oulu, Finland), a moving stage driven by a stepping motor (MRC-999-031, Middleton Spectral Vision, Middleton, WI, USA), a 150 W halogen lamp (A20800, Schott, Southbridge, MA, USA), an InGaAs camera (Goldeye infrared camera: G-032, Allied Vision, Stradtroda, Germany) mounted perpendicular to the sample stage, and a computer with data acquisition and analysis software (FastFrame™ Acquisition Software, Middleton Spectral Vision com., Middleton, WI, USA). To acquire clear images, the parameters of the sample stage speed, the exposure time of the camera, the halogen lamp angle, and the vertical distance between the lens and the sample were set to 10 mms^−1^, 45 ms, 54°, and 12.5 cm, respectively. Samples were placed on the sample stage and captured in line scanning or push broom mode. The acquired hyperspectral images contained 254 wavelength bands stored as a “*.raw” file, along with a header file in the form of an “*.hdr”.

During each scan, one group of 50 millet seeds from each cultivar was arranged on the stage at a time and organized into rows and columns of 5 by 10. The seeds were placed to ensure that no two seeds were touching and that all the seeds were within an area that was about 3 × 7 cm centered on the stage. A total of 10 replicates of 50 millet seeds were collected for each repetition and each cultivar. This makes a total of 5000 seeds scanned for all the 10 cultivars.

### 2.3. HSI Data Preprocessing

After the image acquisition, it was necessary to calibrate the raw hyperspectral images with white and dark reference images to compensate for the effect of illumination, as well as the dark current of the detector. A whiteboard with a reflectance of 99% from a polytetrafluoroethylene (PTFE) Teflon plate was used to acquire the white reference image. Then, lights were turned off and the camera lens was covered completely with a cap to acquire the dark reference image. Then, the hyperspectral images were corrected with the white and dark reference according to the following equation [32]:$$R=\frac{R_{0}-R_{d}}{R_{w}-R_{d}}$$
where *R*_0_ is the raw hyperspectral image, *R_d_* is the dark reference image, and *R_w_* is the white reference image.

### 2.4. Data Analysis

Figure 2 shows the flowchart of the steps for the data analysis followed in this study. First, the seed images were segmented as the region of interest (ROI) from the corrected raw hyperspectral images. To segment the ROI, two masks were used to separate each seed in the image. The first mask was used to remove the background by thresholding (1050 nm), so all the seed boundaries in the image were clearly defined. Then, a connectivity function was used to segment each individual seed from the others in the filtered images. This function counted the connected regions (non-zero connected pixels as individual seeds) for each seed and assigned a different intensity value for every seed. Next, to look at each seed individually, a second mask was created by segmenting out all values that were not equal to the intensity value of the seed of interest.

### 2.5. Spectral Feature Extraction and Preprocessing

Based on the segmentation, the mean spectra for each seed were then determined. A loop was used to find each seed of interest in the image, calculating the spectra for each pixel in the seed. Then, a spectrum was averaged for every pixel, calculating, displaying, and saving the mean spectra for each seed. This ROI and spectra extraction were repeated for every seed in each repetition for each type of millet. Using these mean spectra, the seeds could then be labeled and classified by the difference in the spectra reflectance.

After spectral data extraction, the preprocessing steps of wavelength trimming (cropped at both ends by 6 bands at the beginning and 15 at the end), maximum normalization, Savitzky–Golay smoothing (by moving window with a width of 27, and the second-order polynomial), and standard normal variable (SNV) were performed to remove the noisy wavelengths at the edges of each spectrum, scale data, and compensate the particle size scattering and path length difference effects, respectively [33].

### 2.6. Classification Models

The dataset of the 10 cultivars was then arranged for classification, using the samples as rows and the features and labels as columns. PCA was used to compress the data and reduce the dimensionality of the hypercube. The predictor values were the PCA of the spectral data, and the dependent variables were the cultivars of millet. Different machine learning classification algorithms of SVM (support vector machine), RF (Random Forest), kNN (k-Nearest Neighbors), linear discriminant analysis (LDA), and ensemble methods were performed and compared for their test classification accuracies. The model parameters were optimized with the following settings: an RBF kernel with σ = 1 and C = 1 for SVM; 5 as the number of neighbors and Euclidean as the distance metric for kNN; a Gini criterion for RF; a learning rate of 0.1; 1000 as the number of estimators; and a maximum depth of 10 for Gradient tree boosting. The dataset was split into 80% for training and 20% for testing, with this partitioning process being repeated five times. Throughout each iteration, the model underwent training on the training data and assessment using the corresponding testing data. The performance metric was subsequently computed by averaging the testing accuracy across these five repetitions. This approach yields a more dependable estimation of the model’s accuracy compared to a single dataset partitioning. By averaging the accuracy over multiple repetitions, the influence of random fluctuations was mitigated, ensuring a more consistent evaluation of the model’s generalization performance.

All the data analysis and modeling were performed using the Python 3.7 (Python Software Foundation, https://www.python.org/ (accessed on 1 September 2023)) platform and Jupyter Editor Notebook using open-source libraries of spectral, NumPy, scikit-learning, and matplotlib.

## 3. Results and Discussion

### 3.1. Spectral Characteristics of Proso Millet Seeds

The NIR spectral profiles of the ten proso millet cultivars are shown in Figure 3. The mean spectra for all the cultivars follow a similar curve but with differing reflectance values at certain wavelengths. The distinct differences in the spectral curves of the millet cultivars are evidence that the propensity of obtaining a high level of classification for the seeds using HSI is high. The obvious variations in the spectral profile of the millet seeds could be due to the chemical and physical properties of the samples. The peak absorptions were observed at wavelengths around 920, 1200, 1400, and 1700 nm. Other authors reported similar absorption trends for cereal grains at the same wavelengths [13,33]. The absorption in the 960–980 nm region has been related to the second O–H overtone of water [34]. The absorption band at around 1200 nm has also been reported to be related to the C–H stretching second overtone (–CH3 or –CH2) of carbohydrates [35]. The absorption peaks at around 1440 nm are attributed to the first overtone of O-H stretching of water [36] and the first overtone of the N-H stretching vibrations of protein [14]. Noticeable reflectance differences can be observed between the different varieties across various spectral ranges, including the water absorption range. These differences suggest variations in the chemical properties of the varieties, for example, in the number of water-bound O-H groups.

### 3.2. PCA and Preprocessing

HSI data are highly dimensional, and this creates issues in the analysis and application processes due to the complexity of the data. Often, most of the features are redundant and not significant in building strong classifiers. In this study, the dimensions of the spectra were reduced while retaining all details of the data without losing all the original wavelength information, using principal component analysis (PCA). As shown in Figure 4, the variances explained by the first two principal components PC1 and PC2 were 98% and 2%, respectively. This added up to a cumulative explained variance of 100% in the data. In Figure 4, it can also be observed that the samples from different cultivars are clearly clustered with some overlaps between a few classes.

### 3.3. Machine Learning Classifiers for Millet Cultivars

Table 1 shows the performances of different machine learning classifiers based on the mean spectra for each millet sample for the full-band and PCA processed HSI data. The best classification result, shown in bold in Table 1, was obtained using the Gradient tree boosting classifier, an ensemble method with prediction accuracies of 99.46% and 99.16% for the full dimension and PCA preprocessed data, respectively. The PCA-based model gave almost the same accuracy, while the dimensionality of the raw data was significantly reduced from 254 features to only 2 features. The RF classifier, which is also an ensemble method based on decision tree, also provided good results, with prediction accuracies of 98.92% and 98.90% for the full-band and PCA-processed HSI data, respectively. These results are better than those of Wang et al. [14], who obtained an accuracy of 87.5% in the classification of eight millet varieties. The results of the current study agree with those of Zhu et al. [15], who achieved their highest accuracy of 99.8% by using the ensemble learning model for the classification of the hyperspectral images of ten soybean seed varieties. The ensemble method combines the predictions of multiple classifiers to improve overall performance, robustness, and generalization. It works by aggregating the predictions of several models, mitigating individual weaknesses of the models, which leads to a more accurate and stable prediction.

As shown in Table 1, among all the classifiers, the Gradient tree boosting ensemble model performed the best, and SVM gave the poorest results in a classification of proso millet cultivars. One reason for this could be the fact that a SVM is not suitable for large datasets, and it is especially not effective in multiclass scenarios [37,38]. On the other hand, kNN, which gave high accuracy in this study, needs a large sample size with multiple overlapping samples [39].

Table 2 shows the precision, recall, and F1-scores for the two best classifiers of RF and Gradient tree boosting for each millet cultivar. Not only is the accuracy of RF and Gradient tree boosting high, but their precision, recall, and F1-scores are also very high for all ten cultivars, with none scoring below 0.96. The implication of this is that there was a very limited number of false-positive or false-negative results in a multiclass classification dataset.

### 3.4. Optimal Wavelength Selection

In addition to PCA, feature selection was used to streamline the dimensionality of classification models by selecting the most effective wavelengths from the spectra that account for a significant portion of the variance between classes. This is essential for reducing the computational requirement of the classifier and for quick feedback. A Sequential Forward Selection (SFS) algorithm was applied to obtain the optimal wavelengths. Table 3 shows the results of classification for the Gradient tree boosting model with the selected wavelengths, achieving 98.00%, 98.14%, and 97.60% prediction accuracy for 30, 15, and 5 wavelengths, respectively. This table shows the highest prediction accuracy at 15 wavelengths. While still a high prediction accuracy, the feature selection made the classification accuracy slightly worse for this study, which was not unexpected.

As shown in Table 3, SFS selected 5, 15, and 30 optimum wavelengths out of 254 wavebands in the full spectra, eliminating 98.03%, 94.09%, and 88.18% of the total number of features, respectively. Specifically, the SFS selected 15 optimum wavelengths of 900.17, 903.53, 906.88, 910.24, 913.59, 916.95, 920.30, 923.65, 927.01, 930.36, 933.71, 1004.09, 1540.94, and 1673.58 nm. These selected wavelengths were mostly around the peak absorptions at 920, 1200, 1400, and 1700 nm shown in the spectral curves of the millet seeds (Figure 3). The visualization of the spectra (Figure 1) shows that there are distinctions in the reflectance of each cultivar at these regions. In comparison, the classification performance of feature extraction using PCA surpassed the feature selection using SFS. It could be related to missing information or reduced bands in the SFS method by selecting some wavelengths from full-band spectra and ignoring the remaining features. On the other hand, PCA transforms the features into reduced dimensional space without losing any information. Baek et al. [40] reported that the use of SFS for reducing the dimension of binary classes of diseased and healthy rice kernels obtained accuracies >93% depending on the preprocessing method and classifiers used. However, most of the bands selected were within visible-NIR range of their HSI system. Our study gave a better result with five dimension/wavebands than their results. In their study on the plant phenotyping of four wheat lines, Mogini, Yang, and Marchetto [41] developed multispectral models, applying five feature selection methods that included SFS to obtain 15 bands (from 215) that showed significantly low cross-validation error (CV error). They further identified six bands that are amenable to wheat-line morphological identification with equally low CV error. Their F1 scores were less than 0.79, which were not as high as those obtained in our study. There are slight differences in their study compared to ours, and this may explain the difference in our results. However, these studies and ours reiterate the importance of feature selection for developing a robust multispectral classification model for inline/infield deployment in terms of mitigating the challenges encountered in post-processing, such as complex interpretation and expensive computational cost, with the potential to improve classification accuracy.

## 4. Conclusions

In this study, the potential of using HSI in the SWNIR spectral range (900–1700 nm) was investigated for the classification of ten different proso millet cultivars, namely Cerise, Cope, Earlybird, Huntsman, Minco, Plateau, Rise, Snowbird, Sunrise, and Sunup. Different classification models were built based on the mean spectra for each sample. The best classification performance was obtained from the Gradient tree boosting and RF classifiers that resulted in test dataset classification accuracies of 99.4% and 99.1%, respectively. Using feature selection to reduce the most relevant wavelength gave accuracies of 97.6%, 98.14% and 98%, for 30, 15, and 5 wavelengths, respectively. These results indicate the potential of the HSI in the NIR range as a promising tool to evaluate and classify different proso millet cultivars. The significance of this study lies in its contribution to the development of efficient and reliable methods for proso millet cultivar identification. This capability can have a substantial impact on various aspects of the proso millet industry, such as quality control, breeding programs (by helping with the selection of desirable traits and accelerating the development of new proso millet varieties), and market analysis (by supporting targeted marketing strategies and optimizing proso millet production for specific consumer preferences). Future studies could investigate the classification of cultivars based on additional quality parameters, such as protein or moisture content, using advanced spectral analysis techniques. Additionally, exploring the integration of HSI data with other sensing technologies could lead to even more robust and comprehensive proso millet cultivar identification methods.

## Figures and Tables

**Figure 1 foods-13-01330-f001:**
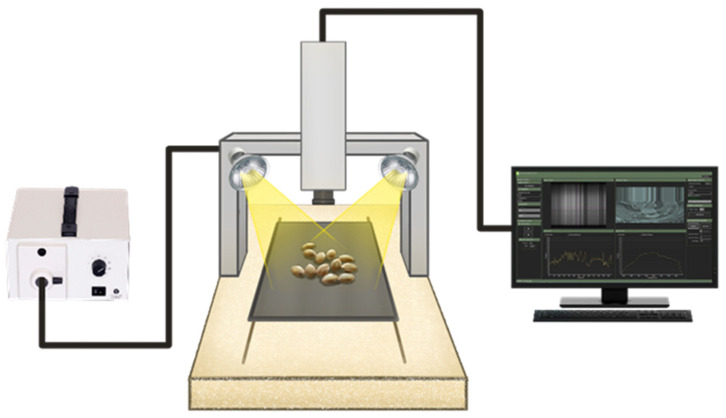
A schematic diagram of the hyperspectral imaging system [31].

**Figure 2 foods-13-01330-f002:**
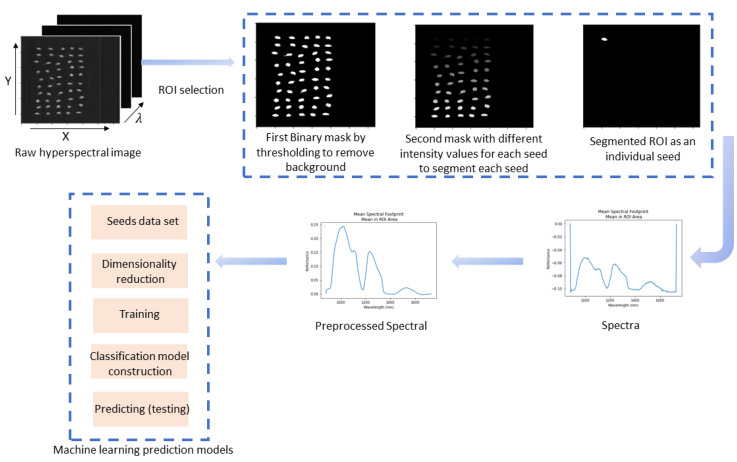
Flow diagram of hyperspectral image data analysis using a machine learning modeling approach.

**Figure 3 foods-13-01330-f003:**
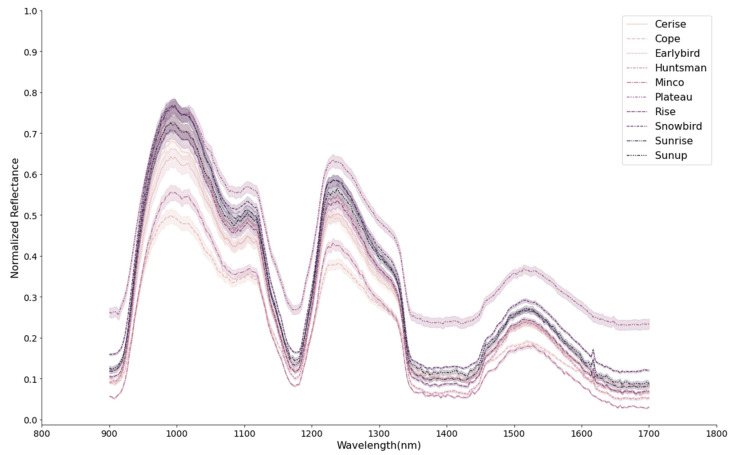
Spectral curves of all millet cultivars.

**Figure 4 foods-13-01330-f004:**
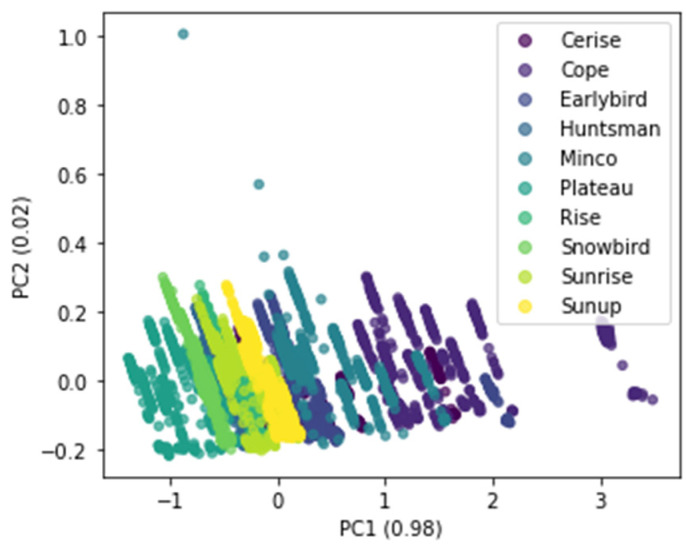
PCA score plots of all ten proso millet cultivars.

**Table 1 foods-13-01330-t001:** Classification accuracy (%) of classifiers for full-band and PCA-processed hyperspectral data.

Classifier ^1^	Full-Band Data *		PCA	
	Training Set	Prediction Set	Training Set	Prediction Set
LDA	100 ± 0.00	99.28 ± 0.20	82.20 ± 0.42	81.64 ± 1.23
SVM	69.40 ± 0.38	68.62 ± 1.42	69.38 ± 0.46	68.62 ± 1.42
kNN	95.88 ± 0.13	93.82 ± 0.46	95.06 ± 0.12	92.74 ± 0.45
RF	100 ± 0.00	98.92 ± 0.33	100 ± 0.00	98.90 ± 0.17
Gradient tree boosting	100 ± 0.00	99.46 ± 0.10	100 ± 0.00	99.16 ± 0.30

^1^ LDA: Linear discriminant analysis; SVM: support vector machine; kNN: k-Nearest Neighbors; RF: Random Forest. Bolded line indicates the best result. * No dimensionality reduction.

**Table 2 foods-13-01330-t002:** Classification performance of Random Forest and Gradient tree boosting models in the classification of proso millet cultivars.

Classifier	Cultivar	Precision	Recall	F1-Score
RF	Cerise	1.00	1.00	1.00
	Cope	1.00	1.00	1.00
	Earlybird	1.00	1.00	1.00
	Huntsman	0.98	0.99	0.99
	Minco	1.00	1.00	1.00
	Plateau	1.00	0.98	0.99
	Rise	0.99	0.98	0.99
	Snowbird	1.00	0.98	0.99
	Sunrise	0.96	0.98	0.97
	Sunup	0.98	1.00	0.99
Gradient tree boosting	Cerise	0.99	1.00	0.99
	Cope	1.00	1.00	1.00
	Earlybird	1.00	1.00	1.00
	Huntsman	0.99	0.99	0.99
	Minco	1.00	1.00	1.00
	Plateau	0.99	0.99	0.99
	Rise	1.00	0.99	1.00
	Snowbird	1.00	0.99	1.00
	Sunrise	1.00	0.98	0.99
	Sunup	0.97	1.00	0.99

**Table 3 foods-13-01330-t003:** Classification performance and selected wavelengths tested using the Sequential Forward Selection (SFS) algorithm for the classification of proso millet cultivars using hyperspectral imaging.

Classifier	No. of Features	Wavebands (nm)	Classification Accuracy
Gradient tree boosting	30	900.17, 903.53, 906.88, 910.24, 913.59, 916.95, 920.30, 923.65, 927.01, 930.36, 933.71, 937.07, 940.42, 943.77, 947.13, 950.48, 953.83, 957.18, 960.53, 963.89, 967.24, 970.59, 973.94, 977.29, 980.64, 983.99, 1004.09, 1540.94, 1673.58	98.00%
	15	900.17, 903.53, 906.88, 910.24, 913.59, 916.95, 920.30, 923.65, 927.01, 930.36, 933.71, 1004.09, 1540.94, 1673.58	98.14%
	5	900.17, 903.53, 1004.09, 1540.94, 1673.58	97.60%

## Data Availability

The data presented in this study are available on request from the corresponding author due to the policy of the funding agency and authors’ institutions.

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
