# Peer review of "Hyperspectral Imaging and Machine Learning as a Nondestructive Method for Proso Millet Seed Detection and Classification"

_foods, 2024, doi:10.3390/foods13091330_

Round 1

Reviewer 1 Report

Comments and Suggestions for Authors

The manuscript is written with clear understanding of the project addressed, but there are some problems. It is recommended to review after major revisions to the manuscript. My specific comments are as follows: 

1. In line 100-155,a literature summary needs to be added to this paragraph.

2. The Introduction section of the manuscript describes the importance of differentiating food composition, purity and physicochemical properties, but the following study is only about differentiating between different varieties of wheat.

3. In line 222, what is the basis for demonstrating that the visible changes in the spectral curve of millet are due to the chemical and physical properties of the sample. It is recommended that this be supported by literature.

4. The preprocessing model used is not specifically described in the article and why such methods were selected as preprocessing methods.

5. The description in the conclusion section is too concise and should specify what the significance of the study is and what impact it will have on the follow-up.

Reviewer 2 Report

Comments and Suggestions for Authors

Dear Authors,

The manuscript with title "Hyperspectral Imaging and Machine Learning as Nondestructive Method for Proso Millet Seed Detection and Classification" is interesting and deals with important area of research which can have application in industry for selection of millet seeds.

However, I suggest improving of manuscript in few main points:

1. The authors should explain more detailed how this modern hyperspectral image data analysis using machine is working regarding the chemical composition of millets seeds, not only morphological.

2.Line 225-230: The authors explained the vibrations in the NIR spectrum from 920 to 1700 nm. Is it possible this modern technique to predict the percentage of protein, oil, fibers and other constituents in millet seeds?

3. Which are the chemical differences between Cerise, Cope, Earlybird, Huntsman, Minco, Plateau, Rise, Snowbird, Sunrise, and Sunup varieties of millet seeds?

4. Which parameters are limited regarding the selection in terms of color, percentage of water and other physico-chemical characteristics of millet seeds?

I suggest major revision of the manuscript.

Reviewer 3 Report

Comments and Suggestions for Authors

The authors should address the following comment.

- Sample preparation. Why remove the hull of the seeds? If the development of a fast sorting system is of interest, the samples much be analyzed intact. Authors should discuss how preparing the sample prior to scanning and predicting the seed cultivar is useful? 

- Why not challenge the model with fully independent data? You have a lot of seeds from each cultivar. Take some out of the calibration set and use them as an independent test set. This would make the article a lot more robust.

- No details are provided for any of the algorithms used. Each algorithm has many parameters that need to be optimized. They should be provided so the study can be replicated.

Figure 3 - there are sharp peaks at 1620 nm for some cultivars. what is it? artifact or reach signal? If so, it should be discussed.

Comments on the Quality of English Language

line 71 - "power machine" - insert "of"

line 80 - "focus" - it should be focusses

line 131 - "deep-reezer" - freezer

Round 2

Reviewer 1 Report

Comments and Suggestions for Authors

Well done!

Reviewer 3 Report

Comments and Suggestions for Authors The article has been adequately improved.